# High-Temperature Quantum Hall Effect in Graphite-Gated Graphene Heterostructure Devices with High Carrier Mobility

**DOI:** 10.3390/nano12213777

**Published:** 2022-10-26

**Authors:** Siyu Zhou, Mengjian Zhu, Qiang Liu, Yang Xiao, Ziru Cui, Chucai Guo

**Affiliations:** College of Advanced Interdisciplinary Studies & Hunan Provincial Key Laboratory of Novel Nano-Optoelectronic Information Materials and Devices, National University of Defense Technology, Changsha 410073, China

**Keywords:** Graphene, hBN, heterostructure, field-effect devices, carrier mobility, quantum Hall effect

## Abstract

Since the discovery of the quantum Hall effect in 1980, it has attracted intense interest in condensed matter physics and has led to a new type of metrological standard by utilizing the resistance quantum. Graphene, a true two-dimensional electron gas material, has demonstrated the half-integer quantum Hall effect and composite-fermion fractional quantum Hall effect due to its unique massless Dirac fermions and ultra-high carrier mobility. Here, we use a monolayer graphene encapsulated with hexagonal boron nitride and few-layer graphite to fabricate micrometer-scale graphene Hall devices. The application of a graphite gate electrode significantly screens the phonon scattering from a conventional SiO_2_/Si substrate, and thus enhances the carrier mobility of graphene. At a low temperature, the carrier mobility of graphene devices can reach 3 × 10^5^ cm^2^/V·s, and at room temperature, the carrier mobility can still exceed 1 × 10^5^ cm^2^/V·s, which is very helpful for the development of high-temperature quantum Hall effects under moderate magnetic fields. At a low temperature of 1.6 K, a series of half-integer quantum Hall plateaus are well-observed in graphene with a magnetic field of 1 T. More importantly, the *ν* = ±2 quantum Hall plateau clearly persists up to 150 K with only a few-tesla magnetic field. These findings show that graphite-gated high-mobility graphene devices hold great potential for high-sensitivity Hall sensors and resistance metrology standards for the new Système International d’unités.

## 1. Introduction

The quantum Hall effect (QHE) is a perfect phenomenon that reflects the microscopic quantum physics behavior in the mesoscopic scale. It has attracted great attention since it was discovered in 1980 [1]. Charged particles moving rapidly along the edge channels are not affected by impurities or disorders, forming a one-dimensional conductive channel with quantum conductivity of $e^{2}/h$. Considering that Landau levels are discrete, each Landau level will form an edge channel, so the number of Landau levels or filling factors that have been filled determines the quantized Hall conductance [2], and the Hall resistance platform at this time corresponds to $(h/e^{2})/n$. Since the quantum Hall resistance is determined by Planck constant *h* and the basic constant *e* of electron charge, it also leads to the establishment of a new resistance measurement standard. At present, the standard is implemented in GaAs/AlGaAs heterostructure devices [3], which usually require the experimental magnetic field strength B ≈ 10 T and measurement under the condition of T ≈ 1.3 K [4,5]. This experimental condition requires low-temperature liquid-helium and a very strong magnetic system, which is very strict for realistic applications. Alternatively, the quantum anomalous Hall effect (QAHE) [6,7,8,9], the quantum case of the anomalous Hall effect [10,11,12], is a transport phenomenon where the Hall resistance is quantized to the von Klitzing constant due to the spontaneous magnetization of a ferromagnetic material even at zero magnetic field. Similar to the QHE under strong magnetic fields, the quantized Hall resistance of QAHE is also supposed to be universal, independent of the experimental details. However, the quantization accuracy of QAHE reported so far is much worse than that of QHE [13,14,15]. In addition, in the present study, all the precision measurements of QAHE were performed at extremely low temperatures below 300 mK [16]. To make the QAHE-based resistance standard more practical, a further increase in the operating temperature would be required.

Recently, a more accurate and practical quantum Hall resistance standard has been rapidly developed by utilizing the unique electronic characteristics of graphene [17,18,19,20,21]. Compared with traditional GaAs/AlGaAs heterostructure devices, graphene devices can observe the QHE at higher temperatures and lower magnetic fields [22]. The reason why graphene can show QHE in a low magnetic field is that the energy interval between the first two Landau levels [21,23] is $∆E_{\mathrm{Graphene}}=36\sqrt{B\;}\mathrm{meV}/T^{1/2}$, which is much larger than $∆E_{\mathrm{GaAs}}=1.7B\;\mathrm{meV}/T$ of traditional GaAs/AlGaAs heterostructure devices [22]. Therefore, the Hall resistance of graphene can be accurately quantified to $h/\left(2e^{2}\right)$ in such a low magnetic field. However, to achieve such accuracy, graphene quantum Hall devices also need to have high carrier mobility. Previous studies have shown that graphene devices with a carrier mobility of 1.5 × 10^4^ cm^2^/V·s can exhibit good QHE at T=4 K and B=14 T, and they discovered the QHE in graphene occurs at a half-integer filling factor [23]. Additionally, chemical vapor deposition (CVD) grown graphene [24] with a high carrier mobility of 6 × 10^5^ cm^2^/V·s shows a fractional quantum Hall effect (FQHE) at T=0.3 K and B=12 T. Room-temperature QHE [25] is early discovered at a very large magnetic field of B=45 T. Therefore, in order to achieve higher performance of graphene quantum Hall devices under relaxed conditions, high carrier mobility is required. Recently, the discovery of other two-dimensional (2D) crystals, such as hexagonal boron nitride (hBN), leads to the emergence of a van der Waals (vdW) heterostructure using a layer-to-layer assemble strategy. The properties of vdW heterostructures can be precisely controlled by adjusting the type of 2D component materials, the number of layers, and the band alignment, which is critical from both fundamental and application points of view. For instance, it has been shown that the hBN encapsulation significantly reduces the scattering in graphene and thus dramatically enhances the carrier mobility of graphene [26,27,28]. The vdW heterostructure that combining hBN and graphene have led to the discovery of many exciting phenomena including Hofstadter’s butterfly, Coulomb drag, and fractional quantum Hall effect, as well as many functional devices such as tunneling field-effect transistors [29,30,31,32,33].

Here, we fabricate a monolayer graphene (MLG) Hall device encapsulated with hBN, the fabrication process is shown in Appendix A. The device uses few-layer graphite (FLG) as gate electrodes and one-dimensional Cr/Au edge electrodes. By utilizing the advanced graphite gate device architecture, the hBN/MLG/hBN vdW heterostructure device demonstrates ultra-high carrier mobility at both room temperature and low temperature, leading to observation of the QHE at an elevated temperature of 150 K and a moderate magnetic field below 8 T. Our results show that for metrology, this may provide a new idea for quantizing Hall resistance benchmark [34]; and for magnetic field detection, graphene also has significant performance advantages in the field of Hall sensors with ultra-high sensitivity [35].

## 2. Results and Discussions

Figure 1a shows the optical image of the device. Our graphene device is made of vdW heterostructure on a silicon substrate as it shows in Figure 1b. The heterostructure is mainly composed of exfoliated MLG, which is encapsulated with hBN as the dielectric layer. The Raman spectra of hBN/MLG/hBN is shown in Figure 1c, the characteristic peak of hBN is at 1366 cm^−1^ and the characteristic G and 2D peak of monolayer graphene is at 1578 cm^−1^ and 2678 cm^−1^, respectively. We notice the G peak intensity of graphene is rather low due to the coverage of hBN, but the 2D peak is symmetrical, which accords with the Raman 2D peak of monolayer graphene. The low charged defect density in hBN can improve the carrier mobility [36], and then FLG is used as the graphite-gated electrode to shield the disorder of charged impurities in the silicon substrate [37], which can also improve the carrier mobility and reduce the inhomogeneity of charge [38,39]. At the same time, in order to reduce the contact resistance between graphene and metal electrode and realize the accurate measurement of the QHE, we use a one-dimensional edge contact method to realize the ability of high-quality electrical contact [28]. Compared with conventional surface contact [40,41,42,43], edge contact can reduce the doping of graphene in the process of metal evaporation [44,45], and can effectively reduce the contact resistance, so as to achieve high electronic performance [46] and make the carrier mobility of graphene devices close to the limit [21].

We first measured the electronic transport properties of the device at room temperature (290 K) and low temperature (1.6 K) by using standard low-frequency locking technology under small ac bias (100 nA) to measure the four-probe longitudinal (*R*_xx_) and Hall resistance (*R*_xy_) and applied the graphite-gate voltage *V*_g_ to adjust the carrier density, the measurement set-up is shown in Appendix A. Figure 2a,d plot the longitudinal resistance of graphene as a function of applied gate voltage (*R*_xx_-*V*_g_) under room temperature (290 K) and low temperature (1.6 K). The gate voltage changes the carrier concentration *n* and thus the resistance of graphene through the capacitance of MLG/hBN/FLG. In consideration of the thickness of the bottom hBN flake, a graphite-gate capacitance per area *A* of $C_{g}/A=\;\varepsilon_{0}\varepsilon_{r}/e\;d_{\mathrm{hBN}}\;\approx {\;6\;\times \;10}^{11}{\;\mathrm{cm}}^{-2}\;V^{-1}$, where $\varepsilon_{0}$ is the dielectric constant, $\varepsilon_{r}\;\approx \;3.9$ is the relative permittivity of hBN, e is the electron charge and $d_{\mathrm{hBN}}=36\;\mathrm{nm}$ is the thickness of the bottom hBN flake [47]. In Figure 2a, the device showed the charge-neutrality point (Dirac point, DP) situated close to zero gate voltage, $V_{\mathrm{DP}}=-0.17\;V$ at room temperature T=290 K. The intensity of the peak in Figure 2a,d represents the largest resistance of graphene at the DP of graphene, while the width of the peak represents the broadening of the DP, mainly caused by the electron-hole puddles in our devices. For graphene near the DP, the concentration of thermally excited carriers Δ*n*_T_ can be estimated as: ${\left(T/\hbar v_{F}\right)}^{2}$, where *T* is temperature, ℏ is reduced Planck constant, and *v*_F_ is Fermi velocity. Therefore, the resistance at DP increases with decreasing temperature, as shown in Figure 2a,d.

The width of the *R*_xx_ peak represents the broadening of the DP, namely, how closely can one approach the DP in graphene. Many theoretical works predict that graphene exhibit $\sigma_{min}=e^{2}/h$ even at vanishing charge density, the DP. However, in a realistic sample charged impurities or structural disorder break up the carrier system into puddles of electrons and holes for *V*_g_ near the *V*_DP_. As a result, the combined (electron plus hole) carrier density in ‘‘dirty’’ graphene never drops below a value Δ*n*, referred to as inhomogeneity density. Both the thermally activated carriers and phonon scattering process increase with increasing temperature, leading to a broadened Δ*n* and width of the *R*_xx_-*V*_g_ peak in Figure 2. In Figure 2a,d, the experimental observation that the peak widths increase and the intensity decrease with the increase of the temperature are in good agreements with previous reports in high-quality graphene devices.

The residual charge carrier fluctuation is $n^{*}=5{.7\times 10}^{10}{\;\mathrm{cm}}^{-2}$ and the charge carrier mobility μ is calculated by using Drude formula σ=neμ, where σ is the electrical conductivity [48]. It can be observed that the mobility of the device can reach around ${1\times 10}^{5}{\;\mathrm{cm}}^{2}/V\;s$, see Figure 2c. Then, we cooled down the device in a ^4^He cryostat system to 1.6 K, the residual charge carrier fluctuation decreases to $1{.7\;\times \;10}^{10}{\;\mathrm{cm}}^{-2}\;\sim \;2{.4\;\times \;10}^{10}{\;\mathrm{cm}}^{-2}$ and the mobility increases to ${3\;\times \;10}^{5}{\;\mathrm{cm}}^{2}/V\;s$ close to the DP, see Figure 2f. Figure 2c,f represent the field-effect carrier mobility as a function of carrier concentration in graphene. The x-axis is the carrier concentration of graphene, which can be tuned from electron side (positive *n*) to hole side (negative *n*) by apply a gate voltage (*V*_g_). The carrier fluctuation (referred as Δ*n*) is determined by the fitting in Figure 2b,d. As shown in Figure 2c,f, the field-effect carrier mobility of graphene is carrier-concentration-dependent. The behavior of non-constant *μ*(*n*) may originate from different scattering efficiency for a particular type of defects in graphene. An alternative explanation consistent with the behavior of *μ*(*n*) is the renormalization of the Fermi velocity *v*_F_, as previously observed in high-mobility suspended graphene devices [49,50,51]. In principle, the valence band (hole doping) and the conduction band (electron doping) are symmetric and the contribution of hole and electron to the carrier mobility is equal in in freestanding perfect graphene sheet. As shown in Figure 2c,f, the extracted carrier mobilities of hole and electron are slight asymmetric, which may originate to the substrate-induced scattering and the electron-hole puddles in graphene. We also compared the device performance of graphene on different substrates, which is shown in Appendix A.

As shown in Figure 3a, under fixed gate voltage (*V*_g_ = 0.5 V) and low temperature (*T* = 1.6 K), the overall positive *R*_xy_ indicates that the contribution is mainly from electrons. Under a low magnetic field, the QHE shows clear Shubnikov-de Haas (SdH) oscillations. With the increase of the magnetic field, *R*_xy_ appears plateaus and *R*_xx_ is vanishing, which is the typical QHE [52,53]. Multiple plateaus are clearly observed in the *R*_xy_ measurements. For the filling factors *ν* = 2 and *ν* = 6, the magnetic fields range from 1.88 T to 2.36 T (*ν* = 2) and from 0.77 T to 0.82 T (*ν* = 6), respectively. Therefore, the *ν* = 2 plateau has the widest range of magnetic field in all the *R*_xy_ plateaus, which can be utilized as the quantum resistance standard. In Figure 3b, we adjust *V*_g_ at a fixed magnetic field (*B* = 1 T) and low temperature (*T* = 1.6 K) to access the quantum Hall plateaus. The lowest point of longitudinal resistance *R*_xx_ corresponds to the plateaus of the Hall resistance, and the quantum Hall plateaus of v=±2, ±6, ±10, ±14, ±18  are clearly observed. The QHE in graphene is different from the conventional QHE because the quantization is turned into a half integer, $R_{\mathrm{xy}}^{-1}=v\left(e^{2}/h\right)$, where the filling factor v=±4 (n+1/2), n is a non-negative integer, the negative filling factor represents the quantum Hall effect (*R*_xy_) in the hole doping regime, while it is positive for electron doping. In Figure 3c, we show the change of longitudinal resistance *R*_xx_ with the regulation of gate voltage (*V*_g_) and magnetic field (*B*), in which we can see the typical fan of Landau levels. The black-colored areas indicate the vanish of *R*_xx_, which means the Landau levels at these areas are clearly visible [54]. At magnetic field *B* > 0.5 T, *R*_xy_(*B*) exhibits plateaus and *R*_xx_ is vanishing, which are the hallmarks of the QHE. At least two well-defined plateaus with values (2*e*^2^/*h*) and (6*e*^2^/*h*), followed by a developing (10*e*^2^/*h*) plateau, are observed before the QHE features transform into SdH oscillations at a lower magnetic field. We observed the equivalent QHE features for holes with negative *R*_xy_ values. Alternatively, we can probe the QHE in both electrons and holes by fixing the magnetic field *B* = 1 T and changing *V*_g_ across the Dirac point. In this case, as *V*_g_ increases, first holes (*V*_g_ < *V*_DP_) and later electrons (*V*_g_ > *V*_DP_) fill successive Landau levels and exhibit the QHE, as shown in Figure 3b. For a fixed magnetic field, the periodicity of *R*_xx_ oscillation in carrier concentration is given by:$$∆n=\alpha ∆V_{g}=4B/\Phi_{0}$$
where $∆V_{g}$ is the periodicity in gate voltage and $\Phi_{0}=h/2e$ is the flux quanta. The factor “4” corresponds to the four-fold spin and valley degeneracy in graphene. 

To further support the high performance of the device, we also measured the transport properties by changing the magnetic field at relatively high temperatures of 100 K and 150 K, and we observe that the device can also show hall plateau values of $h/{\pm 2e}^{2}$. As shown in Figure 4a, at the temperature of 100 K, with the continuous enhancement of the magnetic field, the hall plateau values of $h/{\pm 2e}^{2}$ are gradually obvious. Similarly, at the temperature of 150 K, the same phenomenon can be observed. The reason why the QHE can be observed at such high temperatures is the large cyclotron gaps of Dirac fermions in graphene, which is $\hbar \omega_{c}$. Their energy in the magnetic field is quantified as $E_{N}{=v}_{F}\sqrt{\left|2e\hbar BN\right|},\;$ where $v_{F}\;\approx {\;10}^{6}\;m/s$ is the Fermi velocity and N is the integer of Landau level [21,23]. The energy gap at B=12 T between N=0 and ±1  is ∆E ≈ 1449 K, which means that even with the high temperature of 150 K in our experiment, $\hbar \omega_{c}$ is about 10 times bigger than the thermal energy $k_{B}T$. We also measured at *T* = 200 K and *B* = 12 T, where we can we can barely see the quantum hall plateaus, see Appendix A. The higher the temperature goes, the more the mobility decreases, and we need a much stronger magnetic field to observe the QHE. Moreover, graphene devices have a very high carrier concentration occupied with a single 2D sub-band, which can completely fill the lowest Landau level under a high magnetic field; secondly, our graphene devices still have a high carrier mobility of 1 × 10^5^ cm^2^/V·s at room temperature, which is proportional to scattering time *τ.* Additionally, the scattering time of our device can reach *τ*~10^−12^ s, so the condition of $\omega_{c}\tau =\mu \;\cdot \;B\;\gg \;1$ can be reached at a low magnetic field of several T, compared with the Hall devices based on GaAs/AlGaAs heterostructures [3], HgTe quantum wells [55], and SiO_2_ supported graphene [25], which usually require magnetic fields above 20 Tesla to achieve the quantum Hall effect above the nitrogen temperatures (>77 K). However, we cannot observe the plateaus at 100 K and 150 K with a too low magnetic field (0.1 T), see Appendix A. Generally speaking, in order to eliminate the influence of thermal fluctuation on the measurement, the observation of QHE requires a low temperature. On the one hand, we need to study quantum phenomena under extreme environmental conditions, while on the other hand, the more stable quantum state is also important for the actual measurement under relaxed experimental conditions. 

## 3. Conclusions

In summary, we report high-quality graphene field-effect devices by hBN encapsulation and local graphite bottom gate electrodes. Electrical transport experiments show that the graphene devices exhibit very high carrier mobility in both electron and hole sides. For the electron doping regime, the mobility exceeds 10^5^ cm^2^/V·s and 3 × 10^5^ cm^2^/V·s at room temperature (290 K) and cryogenic temperature (1.6 K). The residual charge carrier fluctuation is only about 10^10^ cm^−2^, indicating the great purity and homogeneity of the graphene devices. The well-defined low carrier concentration in graphene makes it ideal for an ultra-sensitive Hall sensor with a current-related sensitivity of S_I_~ 500 V/(AT) at room temperature, which surpasses both silicon-based Hall sensors and the best Hall sensors based on III/V semiconductors. Further, benefiting from the high carrier mobility, the integer quantum Hall effect is finely developed in graphene under a perpendicular magnetic field below 1 T at low temperature. More surprisingly, clear quantum plateaus at *ν* = 2 in graphene can even persist at an elevated temperature of 150 K, far above the liquid nitrogen temperature. At the same time, the required magnetic field for the high-temperature quantum Hall effect is only about 5 T, which can be realized by commercial superconducting solenoids. The relaxed conditions of quantum Hall effect in graphene, including high temperatures and small magnetic fields, will open new routes for the development of graphene-based resistance standards for the new Système International d’unités.

## Figures and Tables

**Figure 1 nanomaterials-12-03777-f001:**
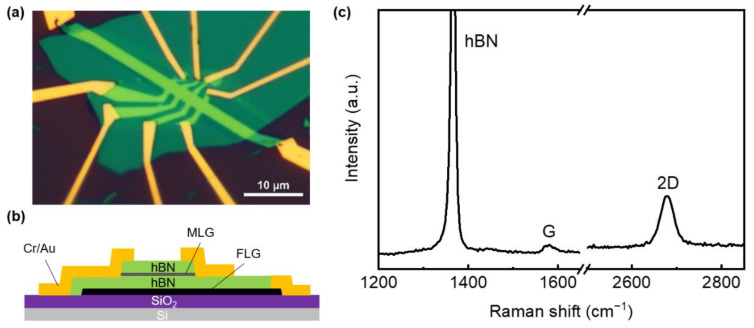
(**a**) Optical microscope image of the graphene quantum Hall device. (**b**) Schematic of the cross-section view of the hBN/MLG/hBN vdW heterostructure device. (**c**) Raman spectra of the hBN/MLG/hBN vdW heterostructure. The wavelength of laser is 532 nm, see Appendix A.

**Figure 2 nanomaterials-12-03777-f002:**
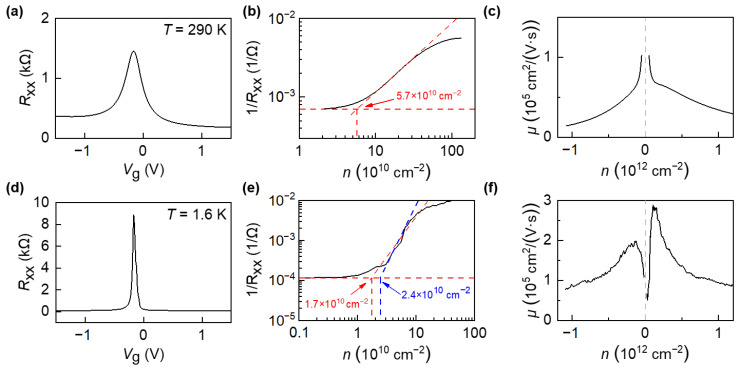
(**a**) Longitudinal resistance *R*_xx_ as a function of gate voltage (*V*_g_) at 290 K under small ac bias current (100 nA). (**b**) The inverse longitudinal resistance *R*_xx_ as a function of the charge carrier density *n*. The residual charge carrier fluctuation is calculated by the intercept of two lines. (**c**) Carrier mobility *μ* as function of charge carrier density *n* for |*n*| > 5.7 × 10^10^ cm^−2^. (**d**,**e**) Same as (**a**,**b**), but measured at *T* = 1.6 K. (**f**) Carrier mobility *μ* as a function of charge carrier density *n* for |*n*| > 1.7 × 10^10^ cm^−^^2^~2.4 × 10^10^ cm^−2^.

**Figure 3 nanomaterials-12-03777-f003:**
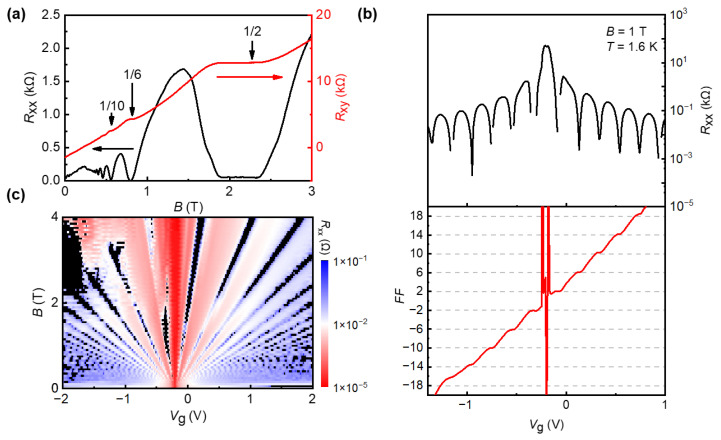
(**a**) Longitudinal resistance *R*_xx_ and Hall resistance *R*_xy_ as a function of magnetic field *B* at *V*_g_ = 0.5 V and *T* = 1.6 K. (**b**) Longitudinal resistance *R*_xx_ (black, top) and filling factor (FF) of *R*_xy_ (red, bottom) as a function of graphite-gate voltage at *T* = 1.6 K, *B* = 1 T. The horizontal dashed gray lines indicate the plateau values of the half-integer QHE. (**c**) False-color map of the longitudinal resistance *R*_xx_ of this device as a function of graphite-gate voltage (*V*_g_) and magnetic field (*B*).

**Figure 4 nanomaterials-12-03777-f004:**
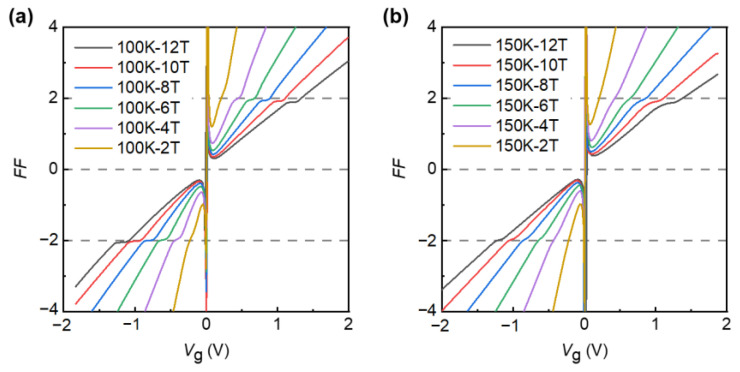
(**a**) Filling factor of *R*_xy_ as a function of graphite-gate voltage at *T* = 100 K in different magnetic field (from *B* = 2 T to 12 T). The horizontal gray lines indicate the hall plateau values of *h*/±2*e*^2^. (**b**) Same as (**a**), but measured at *T* = 150 K.

## Data Availability

The data presented in this study are available on request from the corresponding author.

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
