# Peer review of "High-Temperature Quantum Hall Effect in Graphite-Gated Graphene Heterostructure Devices with High Carrier Mobility"

_nanomaterials, 2022, doi:10.3390/nano12213777_

Round 1

Reviewer 1 Report

Guo et al. report in this manuscript, with title “High-temperature quantum Hall effect in graphite-gated graphene heterostructure devices with high carrier mobility“, on the development of a graphene monolayer, which is encapsulated by means of hexagonal boron nitride, and graphite layer as back gate, to fabricate micrometer-scale graphene-based Hall devices. The authors claim that under a perpendicular magnetic field of a few tesla, the system exhibits good Quantum Hall effect at both low and high temperatures. Overall, the work is well researched and well presented and deserves publication in the journal Nanomaterials. Nevertheless, there are some important issues which must be addressed:

  1. In page 2, line 7, the sentence “Recently, a more accurate...of graphene” is supported by reference 7, which is quite old. It should be revised, as it could be wrong cited.

  2. In page 4, what is exactly the range of the magnetic field in which the “plateaus” of Figure 3(a) are observed?. The authors should indicate this range in the main text.

  3. The conclusions section is too short and should be developed in more detail. Please, check this out.

  4. In the references section, please, complete the list of authors of those references (1-3, 6, 7, 10, 11, 16, … and so on) which are not completed.

Reviewer 2 Report

Submitted manuscript is interesting and the journal selection is correct. Authors make very special efforts in the description of the quantum hall effect in the beginning of the Introduction, which might not be really necessary. However, proper comparison of the anomalous hall effect in disordered ferromagnetic alloys of the transition metals (S.V. Vonsovskii, Magnetism, Nauka, Moscow (1971); Bedyaev et al. Soviet Physics Journal 30(1) 49 – 60 (1987), etc.) and quantum cases would be an advantage as the step of the transition from bulk to nanolevel can be well illustrated here. Another point to add into Introduction is effective medium approximation for extraordinary Hall Effect of ferromagnetic composites (Cohen et al. Physical Review Letters, 30 (15) 696-698 (1973)).

Despite the fact that data for 1.6 K look very interesting it is necessary to add the basis for the physical reasons of the peak (Fig. 2a and d) widths dramatic increase and the intensity decrease with the increase of the temperature. What stratagem can be proposed for the 1.6 K increase? Fig.2e must be discussed from the point of view increased noise – please, show an experimental points with error bars. Why 1.6 K temperature was selected for low temperature test. Is there any physical (as opposed to technological) reason?

The comment “The black colored areas indicate the vanish of Rxx, which means the Landau levels at these areas are clearly visible [31].” Can be even more attractive if Authors discuss the dashed periodic structure related to the disappearance of the longitudinal resistance.

The most interesting data for 0.1 T (Fig. 4) are almost invisible, add the same data with different scale.

The method of the device fabrication seems to be very simple. How repetitive the procedure and the data obtained from one to the other devices fabricated by the same procedure? To what extent and with which accuracy the parameters of the graphene quantum Hall device can be repeated. Could you please to provide optical microscope images of 2-3 graphene quantum Hall devices?

Conclusions are too short to be informative.

Work has significant number of misprints and lost intervals. Not all abbreviations are defined at their first appearance.

Reviewer 3 Report

In this experimental work a graphene device combined with a monolayer graphene encapsulated by a hexagonal boron nitride and a few layers graphite as back gate. It is proved that this device, because of the high carrier mobility, can show a relevant quantum spin Hall effect at both low and high temperature. By using this device as a magnetic sensor, the high carrier mobility leads to a higher sensitivity and to covering a higher range of magnetic fields. More generally, new devices working at high temperatures can be built up and graphene-based resistance standards can be developed. These findings could open the route towards a new idea for quantizing Hall resistance.

In my opinion, the measurements have been carried out very carefully and the results look reasonable and promising. This work advances the field of nanostructures for the novel experiments carried out on the quantum Hall effect and for suggesting a new sensor potentially working in the high-temperature regime.

For the above reasons, this work could match the high standards of the journal but some amendments are necessary.

In particular, the authors should address the following questions/remarks:

1) I think that it should be stressed further in the Introduction and also in the Conclusions the advancement in the field and the novelty made thanks to this work.

2) In the abstract it is stated that the application of graphite gate electrode screens the phonon scattering from SiO2/Si substrate increasing the carrier mobility. However, I didn’t find any discussions in the text about that. The authors should discuss this important point in Section 2. How is the phonon scattering screened from SiO2/Si substrate? Is the increase in the carrier mobility to be considered in graphite, in hexagonal boron nitride which is in direct contact with graphite or in graphene?

3)  Is there a higher contribution of holes or of electrons to the carrier mobility? Or are the two contributions comparable?

4)  Measurements have been done up to T = 150 K and this is considered the high temperature regime in the experiments. What would the authors expect at room temperature? Would there be similar results? I would expect that the phonon scattering is much higher. Would it be screened also at room temperature still leading to a high mobility?

5)  In Figure 2 (c) and (f) it is shown the carrier mobility as a function of charge carrier density for values of |n| larger than some given values. It is not clear to me the real meaning of n in the two panels. Is that an absolute charge carrier or a charge carrier fluctuation? For example, n = 0/cm^2 is less than |n |> 5.7 (1.7) x 10^10/cm^2. What will be the physical implications? By the way, Fig.2 should be replaced by a sharper figure.

6)  It is stated that charge carrier mobility is calculated according to Drude formula linking the mobility to the conductivity. How is the conductivity measured? Or is the conductivity determined from measured resistivity? In that case, how are involved the longitudinal and transverse resistivities?

7) In Figure 3 the label indicating the transverse resistance R_xy should be in red to avoid confusion. Why plateaus are observed for those given values of negative filling fraction? Is there a physical reason why there are these specific values?

8) How is filling factor displayed in Figure 4 and appearing in the expression of transverse resistance defined? Is it related to Landau levels?

9) A brief explanation of what is meant by van der Waals heterostructure would be helpful for a reader.

10)  It is stated that, because of the high mobility, the condition mu B >>1 can be reached at a low magnetic field of several T. How can a low magnetic field be of several T? A clarification of this point is needed.

Minor:

Last paragraph of the Introduction: the word “demonstrate” should read “demonstrates”

Round 2

Reviewer 2 Report

Work was improved up to sufficient level and it can be published in the present state.